# A Little Goes a Long Way: Efficient Long Context Training and Inference with Partial Contexts

**Suyu Ge**[1]\*, **Xihui Lin**[2]\*, **Yunan Zhang**[2]\*, **Jiawei Han**[1], **Hao Peng**[1]
[1]University of Illinois Urbana-Champaign, [2]Microsoft
{suyuge2,haopeng}@illinois.edu

## Abstract

Training and serving long-context large language models (LLMs) incurs substantial overhead. To address this, two critical steps are often required: a pretrained LLM typically undergoes a separate stage for context **length extension** by training on long-context data, followed by architectural modifications to **reduce the overhead of KV cache** during serving. This paper argues that integrating length extension with a GPU-friendly KV cache reduction architecture not only reduces training overhead during length extension, but also achieves better long-context performance. This leads to our proposed LongGen, which finetunes a pretrained LLM into an efficient architecture during length extension. LongGen builds on three key insights: (1) Sparse attention patterns, such as window attention (attending to recent tokens), attention sink (initial ones), and blockwise sparse attention (strided token blocks) are well-suited for building efficient long-context models, primarily due to their GPU-friendly memory access patterns, enabling efficiency gains not just theoretically but in practice as well. (2) It is essential for the model to have direct access to all tokens. A hybrid architecture with 1/3 full attention layers and 2/3 efficient ones achieves a balanced trade-off between efficiency and long-context performance. (3) Lightweight training on 5B long-context data is sufficient to extend the hybrid model's context length from 4K to 128K.

We evaluate LongGen on both Llama-2 7B and Llama-2 70B, demonstrating its effectiveness across different scales. During training with 128K-long contexts, LongGen achieves 1.55x training speedup and reduces wall-clock time by 36%, compared to a full-attention baseline. During inference, LongGen reduces KV cache memory by 62%, achieving 1.67x prefilling speedup and 1.41x decoding speedup. Compared to baselines that apply KV-cache reduction techniques to full-attention long-context LLMs, LongGen achieves substantially stronger performance not only on the Needle-in-a-Haystack retrieval task, but also on more challenging long-context reasoning tasks, including BABILong and RULER.

## 1 Introduction

Transformer-based large language models (LLMs) capable of processing long contexts have unlocked new opportunities across a wide range of applications, such as processing long multi-turn dialogue, understanding code repositories, and answering complex queries that require synthesizing information from multiple documents. However, their quadratic complexity incurs substantial overhead for both training and inference with long contexts (Jiang et al., 2024b; Sun et al., 2024a). To address this, typical approaches involve a two-stage framework (Xiong et al., 2023; Dubey et al., 2024).

- To enhance the model long context capabilities without the substantial overhead of training with long contexts, an LLM is typically pretrained on large short-context datasets (e.g., 2TB of 4K tokens) and undergoes a separate context length extension phase, usually through continual pretraining on a smaller amount of long-context data (e.g., 5B of 128K tokens).

---

\*Authors contributed equally to this research.

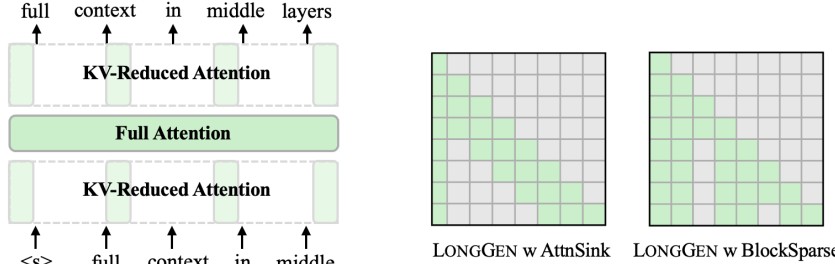

Figure 1: Overview of LONGGEN. **Left:** It uses a hybrid architecture, and applies KV-reduced attention in 2/3 layers at the top and bottom, while keeping the middle 1/3 layers full attention. **Right:** Two KV-reduced attention variants are explored.

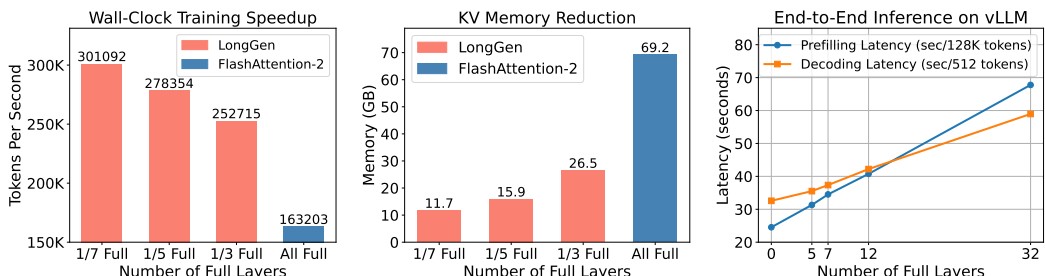

Figure 2: Training and inference efficiency under different sparsity levels. **Left:** Training wall-clock speedup. **Mid:** KV memory reduction. **Right:** Inference speedup. We compare training wall-clock time with FlashAttention and benchmark inference on vLLM. All results are measured on Llama2-7B, which consists of 32 layers in total. "1/7", "1/5", "1/3 Full" and "All Full" indicate using 5, 7, 12, and 32 full layers, respectively.

- Serving a long-context LLM is also challenging due to high memory requirements for caching key and value vectors (KV cache). For example, a 7B Llama-2 model in BF16 precision uses 14 GB for model weights, while the KV cache for a single 128K sequence adds 69 GB - exceeding an H100 GPU's capacity. Recent works have sought to reduce KV cache memory overhead post-hoc by analyzing LLM attention patterns and leveraging these insights to create sparse attention mechanisms, usually without additional fine-tuning.

Despite their promising performance in language modeling perplexity, these methods underperform on long-context retrieval and complex reasoning tasks (§2). This paper argues that integrating the two stages not only builds an efficient long-context LLM with improved long-context performance, but also reduces the cost of length extension training.

Our proposed approach LONGGEN, as illustrated in Figure 1, is a simple and effective hybrid transformer architecture that could be built upon any pretrained transformer to extend its context length. LONGGEN incorporates two key designs: (1) It conducts context extension with various GPU-friendly KV cache-saving designs, including but not limited to window attention (Jiang et al., 2023), attention sink (Xiao et al., 2024), and blockwise strided attention (Zaheer et al., 2020b). Compared with delicately curated KV cache reduction methods, LONGGEN employs a simpler and more efficient KV design, which maintains uniform memory access patterns for attention heads and achieves load balance among token blocks. Practically, our customized triton training kernel inherits from FlashAttention-2 (Dao, 2023), but achieves faster speed in sparse settings, and the inference kernel is well-suited to vLLM (Kwon et al., 2023) for high-throughput serving. (2) LONGGEN uses this sparse attention in a hybrid architecture, where 2/3 of the attention layers use sparse attention while the remaining 1/3 retain full attention, which we find is essential for handling complex tasks requiring direct access to long-context information(§4.4).

Experiments are conducted on a Llama-2-7B base model (Touvron et al., 2023) and its 70B counterpart with group query attention. We extend their context length from 4K to 128K and perform evaluations

on multiple long-context benchmarks, including the needle-in-a-haystack (NIAH) retrieval, and two more distinguishing benchmarks, BABILong (Kuratov et al., 2024) and RULER (Hsieh et al., 2024), where long-context reasoning is also needed. Through ablations on the position and number of sparse attention layers, we study how sparsity level affects long-context capability. Results show that with full layers in the middle, LONGGEN could achieve comparable performances with standard transformers even with 2/3 of its remaining layers being with sparse attention. Meanwhile, LONGGEN only requires lightweight training on 5B Slimpajama (Shen et al., 2023) tokens (less than 0.1% GPU hours of pretraining) and maintains its short context capability, as indicated by the MMLU (Hendrycks et al., 2020) score. With our customized kernel, it achieves a 1.55x wall-clock training speedup, as shown in Figure 2. During inference, it reduces 62% KV cache memory, bringing 1.67x prefilling speedup and 1.41x decoding speedup.

## 2 BACKGROUND

**LLM context length extension.** Due to the overhead of training transformers on long sequences, context extension is usually separated from standard pretraining as a dedicated post-training stage. State-of-the-art models, such as Gemini (Team et al., 2023), Llama series (Touvron et al., 2023; Dubey et al., 2024) and Qwen (Bai et al., 2023) are typically pretrained on large short-sequence corpora, then undergo length extension on relatively smaller amounts of longer sequences. For example, Llama-3 is pre-trained on 15T tokens within an 8K context length, then post-trained on an additional 800B tokens to extend to 128K length (Dubey et al., 2024; Xiong et al., 2023). A related data engineering work (Fu et al., 2024) significantly reduce the tokens needed for context extension to only 5B by carefully balancing the data source to be similar with that of the pretraining corpus. Typically, there is no architectural modifications during length extension. As a result, these long-context models are challenging to serve due to the memory overhead incurred by KV cache (Jiang et al., 2024a). Therefore, techniques are required to reduce the inference-time memory overhead of these models.

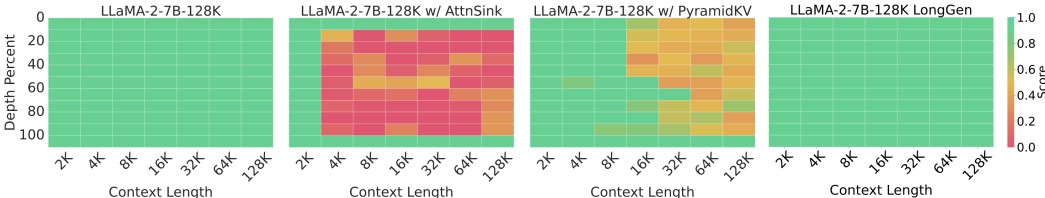

Figure 3: Inference time KV cache reduction methods fail on long context.

**Inference-time KV reduction fails to generalize to long contexts.** To reduce the inference cost of large transformer models, one straightforward idea is to apply inference time KV reduction methods. By only saving the mostly attended KV Cache, they reduce inference memory consumption and forward computing FLOPs. For efficient decoding, the key and value vectors over the context are kept in the GPU memory, usually called KV cache. Its memory overhead has become the primary bottleneck of serving LLMs. Various techniques have been developed to reduce KV cache overhead by storing only a subset of the KV and evicting the rest (Xiao et al., 2024; Liu et al., 2023; Ge et al., 2024). However, as we show here, they often underperform in long-context scenarios, where KV cache reduction is needed the most.

We evaluate two established KV cache reduction methods, Attention Sink (Xiao et al., 2024), and PyramidKV (Zhang et al., 2024a) on a context-extended Llama2-7B-128K model.[1] Attention Sink (AttnSink) only retains initial and local tokens, while PyramidKV identifies the unique attention distribution at each layer and allocates a dynamic KV budget. Similarly to Heavy-Hitter (H2O) (Zhang et al., 2023), PyramidKV preserves tokens that have the largest cumulative attention scores. Additionally, we include the original Llama2-7B-128K baseline and our LONGGEN, and report their performance in Figure 3. For fair comparison, we keep the same KV cache budget for Attn Sink, PyramidKV, and LONGGEN, i.e., 60% of the original full transformer. Figure 3 shows the results on the needle-in-a-haystack (NIAH) retrieval task. Both the original Llama2-7B-128K base model and

---

[1]Details about this Llama2-7B-128K model can be found in §4.1.

LONGGEN can pass the needle retrieval task with 100% accuracy, which we will analyze later in the experiment section §4.2. However, both KV eviction methods underperform on NIAH, especially with > 32K contexts. Intuitively, their underperformance can be attributed to two factors:

- **Lack of full context access.** Due to KV reduction at all layers, the model loses *direct* access to certain positions, making it more challenging for the model to perform accurate retrieval.

- **Lack of sparse context adaption.** The model parameters, especially position embeddings, remain fixed during cache reduction. However, it has been observed that the widely used RoPE-style position embedding (Su et al., 2021) struggles to generalize to unseen position ranges (Wang et al., 2024). After KV reduction, the position range becomes non-contiguous, creating a mismatch with the training setup. This can potentially hurt the model's ability to adapt to longer contexts.

The lessons from the above experiment motivates LONGGEN's key design choices: (1) retaining full context at some layers, and (2) training the model to adapt to sparse attention at the others.

## 3  LONGGEN

LONGGEN improves long-context training and inference efficiency. It finetunes a pretrained LLM into an efficient architecture during length extension. LONGGEN uses a hybrid architecture, with KV-reduced attention patterns in bottom and top layers, while retaining full contexts in the middle layers (Figure 1). We will explore various strategies to initialize the sparse attention, followed by a comprehensive analysis of FLOPs and memory savings to demonstrate the theoretical efficiency benefits of LONGGEN.

### 3.1  CONTEXT EXTENSION WITH EFFICIENT ARCHITECTURES

**A hybrid architecture**    Previous work has found attention is relatively specialized in the middle layers. Some middle-layer attention heads are shown to be strongly related to retrieval and reasoning, indicating refined information aggregation in those layers Wu et al. (2024); Fu (2024); Zhang et al. (2024b). Inspired by their findings, we design LONGGEN to be in an hourglass shape—sparse attention in both ends to efficiently reduce computing cost, and full attention in the middle to maximize long context performance. We will show the necessity of preserving full contexts for middle layers in §4.4. We post-train this hybrid architecture on on longer sequences with the same loss function as in pretraining.

**Different KV-Reduced Attention Strategies**    Efficient attention layers in LONGGEN can be initialized using various KV cache reduction strategies. Instead of choosing attention patterns that adaptively allocate memory for each head, LONGGEN prefers uniformly structured attention patterns. Specifically, we introduce two criteria for sparse attention patterns: (1) *Static access of position.* For each head, the position to be retained should be consistent and agnostic of input tokens. This enables a uniform GPU memory access pattern in training, which allows for optimized sparse training kernel and memory IO. During inference, it is also feasible to enforce the a fixed cache management strategy, thus avoiding extra computation and tensor replacement. From both sides, static attention patterns lead to better efficiency. (2) *Block-wise context handling.*  In GPU, threads are assigned to streaming multiprocessors in block granularity, where each block handles consequent memory space, which is usually significantly larger than the memory occupied by each token. Consequently, token-wise KV design will create idling threads and lower the utilization rate of SRAM. Therefore, we load context by blocks and set the block size to be similar with each thread block size. Guided by the two principles, we equip LONGGEN with two existing KV cache reduction strategies and illustrate them in Figure 1. Our first variant is Attention Sink, which combines initial tokens with local blocks of context window (Xiao et al., 2024). This technique builds upon the proven effectiveness of sliding window attention in long context training, as demonstrated by Mistral (Jiang et al., 2023). An alternative initialization strategy is block sparse attention, which divides the context into multiple blocks and selectively attends to specific block indices based on a predetermined stride (Zaheer et al., 2020a; Qiu et al., 2019).

| | Training FLOPs | KV Cache Mem | Inference Prefilling Time |
|---|---|---|---|
| Full Attention | $\mathcal{O}(L_{\text{full}}(ND^2 + N^2D))$ | $\mathcal{O}(L_{\text{full}}ND)$ | $\mathcal{O}(L_{\text{full}}(N^2D + ND^2))$ |
| Sparse Attention in LONGGEN | $\mathcal{O}(L_{\text{sparse}}(SD^2 + S^2D))$ | $\mathcal{O}(L_{\text{sparse}}SD)$ | $\mathcal{O}(L_{\text{sparse}}(S^2D + SD^2))$ |

Table 1: Efficiency comparison between conventional full attention and the sparse attention in LONGGEN. Training FLOPs count both attention forward and backward operations. $L_{\text{full}}$ and $L_{\text{partial}}$ are the number of full context and partial context layers. $N$ and $D$ represent sequence length and head dimension, respectively. In LONGGEN's attention sink variant, $S$ represents a fixed window block size. In the block sparse variant, $S$ represents a fixed total block size. In both variants, $S$ does not grow with context length, and $S \ll N$.

## 3.2 KERNEL-LEVEL OPTIMIZATION TO IMPROVE EFFICIENCY

Since our implementation differs from traditional transformers in sparse attention computation only, LONGGEN can benefit from the tools developed to improve transformer efficiency, such as flash attention. In fact, we build our customized kernel upon flash attention. Similarly to FlashAttention (Dao et al., 2022), LONGGEN takes an attention mask matrix as input. The attention mask is represented in compressed sparse row format, which consumes $O(N)$ memory instead of the original $O(N^2)$. Note that the sparse mask is static for all heads, so there is no overhead in building the dynamic masks. During forward and backward steps, we skip computing a position by avoiding loading it into HBM if its mask value is zero. This implementation enables FLOPs saving, significantly accelerating training and inference.

During inference, the bottleneck is to load KV cache as fast as possible. Instead of loading the entire head dimension, we split the loading of the head dimension across different thread blocks. We empirically found this to be more efficient than the original FlashAttention, probably because of reduced SRAM usage per loading cycle. For the attention sink initialization, we follow Mistral to use the rolling buffer cache for local sliding window tokens (Jiang et al., 2023).

## 3.3 THEORETICAL TRAINING AND INFERENCE ADVANTAGE

**Training** Since each token needs to be stored to calculate the loss function, LONGGEN does not provide any memory savings during the training process. The major efficiency gain comes from a substantial reduction in training FLOPs. Table 1 demonstrates that when the context length $N$ significantly exceeds the block size $S$, the FLOPs required for sparse attention layers become negligible ($S \ll N$). Consequently, LONGGEN effectively reduces the total FLOPs to a fraction of $L_{\text{full}}/(L_{\text{full}} + L_{\text{sparse}})$ compared to the original transformer model. Empirical studies indicate that a ratio of $L_{\text{full}} : L_{\text{sparse}} = 1 : 2$ provides an optimal balance, achieving significant efficiency gains without compromising the model's long-context performance.

**Inference** During the inference phase, LONGGEN demonstrates notable improvements in both prefilling time and KV cache memory usage, as shown in the last two columns of Table 1. In models with full layers, KV cache typically grows linearly with sequence length, while prefilling time exhibits quadratic growth. However, LONGGEN effectively mitigates these factors. Analogous to the training phase analysis, when $S \ll N$, the cost induced by sparse attention layers becomes negligible, reducing both metrics to $L_{\text{full}}/(L_{\text{full}} + L_{\text{sparse}})$ of the original values.

With our customized triton kernels, LONGGEN can translate the theoretical efficiency gain to wall-clock speedup and GPU memory saving, which we will further illustrate in § 4.3.

## 4 EXPERIMENTS

### 4.1 EXPERIMENTAL SETTINGS

In order to evaluate LONGGEN across different model scales and architecture, we experiment with Llama-2-7B-base (regular multihead attention) and Llama-2-70B-base (grouped-query attention). Following Fu et al. (2024), we continue pretraining them on the same SlimPajama data blend, an open-

| KV Cache Eviction Methods | Attn Sink | H2O | RazorAttention | PyramidKV |
|---|---|---|---|---|
| NIAH Pass Rate | 28% | 32% | 46% | 51% |
| **Long-context Training Methods** | Sliding Window | YOCO | LONGGEN w/ AttnSink | LONGGEN w/ BlockSparse |
| NIAH Pass Rate | 48% | 88% | **100%** | **100%** |

Table 2: Averaged pass rate of different methods on the needle-in-a-haystack retrieval task. All methods are based on a Llama2-7B model. Evaluation context length ranges from 512 to 128K.

| | **Full Attention** | | | | | | | | | | | | | | |
|---|---|---|---|---|---|---|---|---|---|---|---|---|---|---|---|
| | 512 | 1k | 2k | 4k | 8k | 16k | 32k | 64k | 70k | 80k | 90k | 100k | 110k | 120k | 128k | AVG |
| Full | 0.39 | 0.37 | 0.36 | 0.37 | 0.34 | 0.36 | 0.3 | 0.26 | 0.24 | 0.25 | 0.2 | 0.21 | 0.22 | 0.22 | 0.19 | 0.29 |
| | **KV Cache Eviction Methods** | | | | | | | | | | | | | | | |
| Attn Sink | 0.36 | 0.34 | 0.34 | 0.32 | 0.28 | 0.22 | 0.15 | 0.11 | 0.08 | 0.06 | 0.07 | 0.06 | 0.07 | 0.05 | 0.05 | 0.17 |
| H2O | 0.37 | 0.36 | 0.35 | 0.36 | 0.28 | 0.24 | 0.15 | 0.13 | 0.09 | 0.07 | 0.06 | 0.06 | 0.07 | 0.06 | 0.05 | 0.18 |
| RazorAttention | 0.37 | 0.35 | 0.35 | 0.35 | 0.29 | 0.25 | 0.2 | 0.17 | 0.16 | 0.13 | 0.12 | 0.1 | 0.09 | 0.06 | 0.07 | 0.2 |
| PyramidKV | 0.38 | 0.36 | 0.35 | 0.36 | 0.28 | 0.26 | 0.21 | 0.19 | 0.16 | 0.15 | 0.1 | 0.12 | 0.07 | 0.08 | 0.08 | 0.21 |
| | **Long Context Training Methods** | | | | | | | | | | | | | | | |
| Sliding Window | 0.36 | 0.34 | 0.33 | 0.34 | 0.31 | 0.26 | 0.19 | 0.13 | 0.13 | 0.11 | 0.1 | 0.11 | 0.09 | 0.09 | 0.08 | 0.2 |
| YOCO | 0.37 | 0.36 | 0.37 | 0.33 | 0.28 | 0.24 | 0.22 | 0.19 | 0.17 | 0.18 | 0.19 | 0.16 | 0.18 | 0.14 | 0.14 | 0.23 |
| LONGGEN w AttnSink | 0.41 | 0.38 | 0.36 | 0.37 | 0.3 | 0.3 | 0.25 | 0.23 | 0.22 | 0.2 | 0.22 | 0.2 | 0.2 | 0.18 | 0.17 | 0.27 |
| LONGGEN w BlockSparse | 0.39 | 0.36 | 0.36 | 0.34 | 0.31 | 0.3 | 0.25 | 0.24 | 0.23 | 0.21 | 0.21 | 0.21 | 0.22 | 0.2 | 0.18 | 0.27 |

Table 3: Evaluation results on the first five tasks of BABILong. Columns correspond to sequence lengths, rows denote models. Each number indicates the average accuracy of the model over 5 tasks at a given sequence length, calculated over 1000 samples.

source reproduction of Llama-2's pretraining corpus (Shen et al., 2023). We curate long sequence training data by concatenating short texts in Slimpajama to 128K length and marking the document boundary with <bos> token. We train models on 5B tokens, which only account for 0.2% of its 2.4T pretraining corpus. With 32 Nvidia A100-80G GPUs, the post-training of 7B and 70B models with full attention consumes 74 and 387 GPU hours, respectively. The learning rate is 2e-5, the global batch size is 32, and we modify the RoPE base to 5M. Each kernel block handles 64 tokens. For LONGGEN with attention sink, we retain the first block for sink tokens and the most recent 32 blocks for local context. Similarly, for LONGGEN with block sparse, we set the stride length as 64 blocks, meaning we only retain one context block for every 64 blocks. This ensures that when having the same numbers of sparse layers, LONGGEN w/ AttnSink and LONGGEN w/ BlockSparse share similar KV cache budgets, which are estimated as 2K retained tokens for a 128K sequence length. We run inference on the vLLM benchmark [2] and compare efficiency accordingly.

We evaluate on three long-context benchmarks—Needle-in-a-Haystack (NIAH) retrieval, BABI-Long (Kuratov et al., 2024), and RULER (Hsieh et al., 2024). The latter two require models to not only retireve multiple pieces of evidence from the long context, but also reason over them; each evidence sentence is randomly placed in a long sequence. Following BABILong paper, we used 3 different random seeds to generate 2250 test samples for each task, and average model performance on the first five tasks. For RULER, we only evaluate models on its two QA subtasks for two reasons: (1) We observe huge performance variance on multi-needle retrieval and other synthetic tasks. When varying random seeds, the performance can fluctuate up to 40%. (2) Compared with other synthetic counting tasks, the two multi-hop QA tasks are more practical and close to real-world application. For short-context evaluation, we followed Llama2 to test on MMLU (Hendrycks et al., 2020), Math (Hendrycks et al., 2021), and BigBenchHard(BBH) (Suzgun et al., 2022).

---

[2]vllm/benchmark/benchmark_latebcg.py

|                     | NIAH | BABILong | RULER | MMLU  | Math | BBH  |
|---------------------|------|----------|-------|-------|------|------|
| Full Attention-7B   | 100% | 0.29     | 0.41  | 0.419 | 0.12 | 0.32 |
| LONGGEN-7B          | 100% | 0.27     | 0.38  | 0.415 | 0.13 | 0.31 |
| Full Attention-70B  | 100% | 0.46     | 0.67  | 0.661 | 0.32 | 0.49 |
| LONGGEN-70B         | 100% | 0.46     | 0.65  | 0.658 | 0.32 | 0.49 |

Table 4: Overall long context performance of LONGGEN compared with full attention baseline. Evaluation context length ranges from 512 to 128K. NIAH measures the averaged correct answer rate of the models on the needle-in-a-haystack retrieval task. BABILong refers to the averaged accuracy score on the first five tasks on the BABILong benchmark. RULER reports the averaged accuracy on two multi-hop question answering subtasks from RULER.

## 4.2 OVERALL PERFORMANCE

**Baselines** We evaluate LONGGEN against various baselines, reporting performance on NIAH and BABILong datasets in Tables 2 and 3, respectively. The comparative methods span three categories: (1) *Full Attention*: Serving as an upper-bound reference, this approach lacks efficiency optimizations and is computationally expensive for long contexts. (2) *KV Cache Eviction Methods*: We incorporate two widely adopted inference-time KV reduction techniques: Attention Sink (Xiao et al., 2024) and Heavy Hitter (H2O) (Zhang et al., 2023). Additionally, we include recently proposed methods such as RazorAttention (Tang et al., 2024) and PyramidKV (Zhang et al., 2024a), which capture fine-grained attention distribution patterns and adaptively allocate KV budgets. (3) *Long Context Training Methods*: This category includes sliding window attention, YOCO (Sun et al., 2024b), and our proposed approach with both patterns. Sliding window attention at all layers is employed by Mistral for long context training, while YOCO utilizes a cross-decoder design to reuse the KV cache from earlier layers. To ensure a fair comparison, we maintain consistent KV cache budgets across all methods by adjusting the corresponding hyperparameters. Due to variations in implementation, KV budgets may fluctuate by approximately 5% between different methods.

**Performance Evaluation on Long-Context** Analysis of the NIAH results in Table 2 yields two key findings: (1) Long context training methods consistently outperform inference-time KV eviction techniques. As discussed in §2, ensuring full access to the context window and training models to adapt to longer context lengths is crucial. (2) Among the long context training methods, LONGGEN uniquely achieves perfect needle retrieval results. In contrast to window attention, LONGGEN retains the entire sequence. Compared to YOCO, LONGGEN more closely resembles the original transformer architecture in pretraining. Furthermore, our method takes advantage of the information aggregation capabilities in middle layers.

To assess advanced retrieval and reasoning capabilities, we present results on BABILong in Table 3, detailing accuracy at each sequence length. While exhibiting similar trends to NIAH, BABILong provides better differentiation between methods. Among efficient models, LONGGEN demonstrates superior performance, achieving an average accuracy of 0.27. Notably, despite achieving 100% accuracy on NIAH, LONGGEN still slightly underperforms full attention on more challenging tasks. These findings underscore the importance of evaluating models on challenging reasoning tasks.

**Comprehensive Comparison with Full Attention** We compare LONGGEN-7B and LONGGEN-70B with their full attention counterparts, which serves as the upper bound for our hybrid sparse architecture. Our experiments revealed no significant performance disparities between LONGGEN w/ AttnSink and LONGGEN w/ BlockSparse since their accuracies exhibited minor fluctuations across different training iterations. For simplicity, we present the evaluation results of LONGGEN w/ AttnSink on both short-context and long-context benchmarks in Table 4. It is worth noting that we do not express a preference for either strategy in terms of performance.

Although our proposed models demonstrate marginally lower performance compared to the full attention models, they achieve comparable results across various context scales while offering substantial efficiency improvements. Our analysis yielded two key findings: (1) Their performance gap is more significant in long-context scenarios compared to short-context ones. This can be

| Full Layers Position | 512 | 1k | 2k | 4k | 8k | 16k | 32k | 64k | 70k | 80k | 90k | 100k | 110k | 120k | 128k | AVG |
|---|---|---|---|---|---|---|---|---|---|---|---|---|---|---|---|---|
| Top | 0.39 | 0.38 | 0.35 | 0.31 | 0.26 | 0.22 | 0.16 | 0.08 | 0.12 | 0.09 | 0.09 | 0.09 | 0.08 | 0.08 | 0.07 | 0.18 |
| **Middle (LONGGEN)** | 0.41 | 0.38 | 0.36 | 0.37 | 0.3 | 0.3 | 0.25 | 0.23 | 0.22 | 0.2 | 0.22 | 0.2 | 0.2 | 0.18 | 0.17 | 0.27 |
| Bottom | 0.38 | 0.35 | 0.32 | 0.27 | 0.25 | 0.22 | 0.2 | 0.14 | 0.14 | 0.13 | 0.11 | 0.12 | 0.11 | 0.13 | 0.11 | 0.2 |
| Interleaving | 0.35 | 0.33 | 0.33 | 0.31 | 0.29 | 0.3 | 0.25 | 0.2 | 0.21 | 0.18 | 0.18 | 0.19 | 0.18 | 0.16 | 0.15 | 0.24 |

Table 5: Ablation on the placement of the full attention layers. We train the base Llama2-7B model with 32 layers in total on 128K context length, and report results on BABILong. We fix the full attention layer budget as one-third of all layers (12 sparse layers in total) and test varied locations of full layers. Top, middle, and bottom settings stack all full layers together while interleaving setting inserts every full layer between two sparse ones.

| Full Layers Num | 512 | 1k | 2k | 4k | 8k | 16k | 32k | 64k | 70k | 80k | 90k | 100k | 110k | 120k | 128k | AVG |
|---|---|---|---|---|---|---|---|---|---|---|---|---|---|---|---|---|
| All Full | 0.39 | 0.37 | 0.36 | 0.37 | 0.34 | 0.36 | 0.3 | 0.26 | 0.24 | 0.25 | 0.2 | 0.21 | 0.22 | 0.22 | 0.19 | 0.29 |
| **1/3 Full** | 0.41 | 0.38 | 0.36 | 0.37 | 0.3 | 0.3 | 0.25 | 0.23 | 0.22 | 0.2 | 0.22 | 0.2 | 0.2 | 0.18 | 0.17 | 0.27 |
| 1/5 Full | 0.39 | 0.37 | 0.36 | 0.36 | 0.31 | 0.29 | 0.24 | 0.22 | 0.21 | 0.21 | 0.2 | 0.18 | 0.17 | 0.17 | 0.15 | 0.26 |
| 1/16 Full | 0.38 | 0.37 | 0.34 | 0.33 | 0.32 | 0.27 | 0.21 | 0.15 | 0.14 | 0.12 | 0.11 | 0.13 | 0.12 | 0.11 | 0.09 | 0.21 |

Table 6: Ablation on the number of full layers in LONGGEN w/ AttnSink. The base model is Llama2-7B-128K with 32 layers in total. We locate all full attention layers in the middle of the model for the best performance. All, 1/3, 1/5, and 1/16 Full indicate using 32, 12, 6, and 2 full layers, respectively.

attributed to the fact that most short context is fully captured by the local blocks in LONGGEN, whereas information from longer distances is only sparsely represented. (2) The performance gap between ours and the upper bound narrows as the model scale increases. This trend may be attributed to the higher degree of information sparsity exhibited by larger models, thus highlighting the potential for applying our approach to models with even larger parameter counts. These findings underscore LONGGEN's potential for scaling to more expansive parameter spaces.

## 4.3 EFFICIENCY IMPROVEMENT

**Training** Similar to FlashAttention (Dao et al., 2022), we use wall-clock training speedup to evaluate training efficiency. We measured the latency of training a Llama2-7b model on 128K sequence length with 256 A100-80G GPUS. We set tensor parallel size as 8 and use distributed optimizer and checkpoint activation. Results are presented in the leftmost part of Figure 2. By adding sparse layers, LONGGEN significantly improves training throughput. Our chosen "1/3 Full" setting could achieve 1.55 times speedup in total wall-clock training time. While increasing the sparsity level will bring us more efficiency gain, we avoid doing this in pursuit of optimal long-context performance. We will explain more details in §4.4.

**Inference** To verify that LONGGEN will bring practical system-level improvement during serving, we pair our inference kernel to be compatible with vLLM. Using vLLM's official benchmarking tools, we report KV cache memory saving and inference latency reduction in Figure 2. Both memory and latency are measured on one single 128K sequence with a tensor parallel size of 4. From the figure, our chosen "1/3 Full (12 Full layers)" setting reduces memory consumption from 69.2 GB to 26.5 GB (a 62% reduction). Meanwhile, it saves 40% profiling time and 29 % decoding time. The results demonstrates LONGGEN's potential for efficient long-context serving. Similarly, we do not opt for sparser architecture due to performance consideration.

## 4.4 IDENTIFYING ESSENTIAL FACTORS FOR LONG-CONTEXT

In this section, we investigate the impact of varying the number and location of full attention layers on long-context performance.

**Position of Full Layers**   We first examine the effect of full layer positioning by maintaining a constant number of 12 full layers and altering their placement. We explore four configurations: (1) stacking in the top layers, (2) stacking in the middle layers, (3) stacking in the bottom layers, and (4) interleaving in the middle layers. Table 5 presents the results on the BABILong dataset, demonstrating that the position of full layers significantly influences long-context performance. Notably, positioning full layers in the middle of the network yields optimal results. These findings align with previous research indicating that attention heads in the middle layers play a crucial role in information aggregation.

**Number of Full Layers**   Subsequently, we investigate the relationship between the number of full layers and model performance. While increasing the number of full layers is expected to approach the performance upper bound, we aim to identify an optimal balance between computational efficiency and accuracy. Table 6 illustrates the performance for various numbers of full layers. Our analysis reveals that maintaining full layers between 1/5 (6 layers) and 1/3 (12 layers) of the total network depth adequately preserves satisfactory long-context accuracy.

These empirical observations inspire the final architectural design of LONGGEN, which incorporates 1/3 full layers positioned in the middle of the network.

## 5   RELATED WORK

### 5.1   EFFICIENT LONG CONTEXT TRAINING ARCHITECTURE

To overcome the quadratic complexity of transformers, several model architecture alternatives have been introduced. With recent key modifications (Gu & Dao, 2023) to enhance training efficiency, state-space models emerged as they only require linear computation complexity. Gu et al. (2020; 2021) first mitigate the training scalability issue of sequential models with hippo matrix and parallelizable operands. Mamba (Gu & Dao, 2023; Dao & Gu, 2024) introduces input-dependent state parametrization to SSMs and optimized hardware-aware implementations. (Park et al., 2024; Zuo et al., 2022; Ma et al., 2023; Ren et al., 2023; Glorioso et al., 2024; Ren et al., 2024) propose hybrid architectures by integrating different variations of transformer layers into SSMs. However, as recent studies suggests, SSMs are less competitive in long context capabilities compared to transformers (Wen et al., 2024; Arora et al., 2024). Earlier, several works attempts to inject sparsity into transformers to improve efficiency in long context settings (Tay et al., 2023; Child et al., 2019; Beltagy et al., 2020; Zaheer et al., 2020a). However, as pointed out in Dao et al. (2022), these algorithms can hardly bring wall-clock speed-up due to poor compatibility with accelerators. Meanwhile, Sun et al. (2024b) revives the encoder-decoder architecture to save KV cache, where all decoder layers consume the same KV vectors from the encoder side with cross attention. However, it's unclear how to use the architecture in context extension, as it diverges significantly from default decoder-only models.

### 5.2   INFERENCE TIME KV CACHE REDUCTION

Recently, many attempts have been made to evict KV cache at inference time for faster long context serving scenarios (Miao et al., 2023). Zhang et al. (2023); Liu et al. (2023) proposes to discard less important KV vectors based on the accumulated attention score. Han et al. (2023); Xiao et al. (2024) find LLMs tend to store important global information into initial tokens, known as attention sink. By keeping only the attention sink and recent tokens in cache, LLMs can maintain a reasonably well performance. Ge et al. (2024); Li et al. (2024) design algorithms to adaptively maintain KV cache with hybrid eviction policies. PyramidKV (Zhang et al., 2024b) find that the sparsity in attention varies between layers and proposes to dynamically allocate KV cache budget for each layer. This line of work shows that LLMs can work with partial context at inference time. (Li et al., 2024; Zhang et al., 2024b) However, Jiang et al. (2024a); Zhang et al. (2024b) shows that many of the methods will have significant degradation in long context tasks. Also, it's questionable whether complex eviction methods are compatible with realistic serving systems, as calculating accumulative attention score and releasing arbitrary memories are challenging to implement in PagedAttention and Prefix Caching (Kwon et al., 2023; Zheng et al., 2023).

# 6 CONCLUSION

In this paper, we propose LONGGEN to finetune a pretrained LLM into an efficient architecture during length extension. We incorporate GPU-friendly sparse attention to enable practical efficiency gain. Additionally, we find using a hybrid architecture with 1/3 full attention layers and 2/3 efficient ones can achieve a balanced trade-off between efficiency and long-context performance. Through lightweight training, we extend the context length of llama2-7B and 70B from 4K to 128K. Evaluations on multiple long-context benchmarks suggest that LONGGEN reaches on-par performance with full attention on long-context retrieval and reasoning. While at the same time, it reduces wall-clock training time by 36% and KV-cache by 62%, reaching 1.67x acceleration on prefilling and 1.41x on decoding stage. Our study motivates future work on efficient transformer architectures and low-cost methods for long context extension.

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
