# OpenReview forum: "A Little Goes a Long Way: Efficient Long Context Training and Inference with Partial Contexts"
_ICLR.cc/2025/Conference — ICLR 2025 Poster_

### Official Review · Reviewer_83A1 · 2024-10-30

**Soundness:** 3
**Presentation:** 3
**Contribution:** 2
**Rating:** 6
**Confidence:** 4

**Summary:**

The authors finetunes a pretrained LLM into an hybrid architecture that consists 1/3 full attention layers and 2/3 sparse attention layers. By incorporating full attention layers, LongGen allows the model to access to certain positions directly, enhancing the performance on accurate retrieval tasks. Experimental results show that LongGen incurs no loss on the needle-in-a-haystack retrieval task and maintains model performance on tasks with short context, such as MMLU, demonstrating its effectiveness.

**Strengths:**

- The work highlights the importance of including full attention layers for models to achieve accurate retrieval.
- The paper is well-written.
- Experimental results indicate that LongGen accelerates both training and inference for long context while preserving model performance.

**Weaknesses:**

- Since training on long context constitutes a small portion of pre-training, the training speedup of LongGen is limited during pre-training.
- LongGen with AtteSink and BlockSparse demonstrates similar performance, necessitating a detailed explanation for this observation.

**Questions:**

See weakness

---

> ### Comment · Reviewer_83A1 · 2024-11-27
>
> After reading the comments from reviewers gkd86 and GZk8, I concur that this is not the first paper to propose post-training with a sparse attention mechanism, which limits the novelty of the contribution. Additionally, I find it concerning that the authors have not responded to these comments. Based on these concerns, I have slightly lowered my score but still recommend acceptance.

---

> > ### Author Response · Authors · 2024-12-01
> > **Response to Reviewer 83A1's Comment**
> >
> > **Additional1: After reading the comments from reviewers gkd86 and GZk8, I concur that this is not the first paper to propose post-training with a sparse attention mechanism, which limits the novelty of the contribution.**
> >
> > Thanks for pointing out that  KV-reduced attention and lightweight long-context training have received a growing amount of recent interest. This paper aims to answer three research questions that remain unclear in existing works: **1) Whether KV-reduced attention could lead to end-to-end speedup.** This would require the attention pattern to be hardware-aware, supported by a customized training or inference kernel. Besides, some KV-reduced attention methods are incompatible with vllm and TensorRT, the two most popular efficient inference engines. Given this, it is hard for previous KV-reduced attention to bring end-to-end system speedup. To better illustrate this, we compare the throughput of Attention-sink (pytorch-implementation), vllm (full attention), and LongGen (vLLM compatible) in the table below.
> >
> > | Model                                      | Throughput (token/second) | Improvement |
> > |--------------------------------------------|---------------------------|-------------|
> > | Attention-sink (pytorch-implementation)    | 23.5                      | x1          |
> > | vLLM (full attention)                      | 63.3                      | x2.7        |
> > | LongGen (vLLM compatible)                  | 97.6                      | x4.2        |
> >
> > From the table, we can observe that being vLLM-compatible enables LongGen to improve throughput upon paged attention. **2) Whether KV-reduced attention remains effective in long-context scenarios.** In Figure 3 of the paper, we find directly applying KV-reduced attention to long contexts does not work well. Motivated by the drawbacks, we design LongGen to extend this line of research to long context. **3) Whether lightweight training is still enough for length extension with KV-reduced attention.** Since KV-reduced attention changes the model architecture, it is unclear whether more computing should be put into adapting the model to the new sparse architecture. Our study verifies that 5B tokens suffice the need.
> > Therefore, we believe **LongGen is novel and it provides valuable contributions in terms of 1) combining KV-reduced attention and context length extension; 2) a highly efficient training and inference kernel.**
> >
> > **Additional2: Additionally, I find it concerning that the authors have not responded to these comments. Based on these concerns, I have slightly lowered my score but still recommend acceptance.**
> >
> > We apologize for replying late. We spent some time 1) benchmarking efficiency at different lengths using vLLM; and 2) reproducing several KV cache methods, which other reviewers mention. And we prefer to reply to all reviewers simultaneously to show respect. We hope our replies answer your questions. Please feel free to ask if you have any other questions or concerns.

---

> > > ### Comment · Reviewer_83A1 · 2024-12-02
> > >
> > > Thanks for your detailed responses. Most of my concerns have been addressed. I will maintain my score (still recommend acceptance).

---

> ### Author Response · Authors · 2024-12-01
> **Response**
>
> We thank reviewer 83A1 for the valuable time and constructive feedback. Here are our responses to the questions.
>
> **W1: Since training on long context constitutes a small portion of pre-training, the training speedup of LongGen is limited during pre-training.**
>
> We agree that long context training only accounts for a small portion of pre-training. However, LongGen could potentially also be used in the pretraining stage, e.g., pretraining a model with sparse attention. In this paper, we do not explore pretraining due to resource limits. However, when applied to the pretraining stage, our kernel can bring the same level of efficiency improvement. Besides, the training kernel is highly configurable and allows different sparsity levels, which could be a convenient tool for studying the impact of sparsity on pretraining. We hope LongGen can motivate others with more powerful computational resources to study this problem. Upon release, others could use our kernel to conduct pretraining experiments. It would be interesting to investigate the performance and speed trade-off on pretraining with sparse attention.
>
> Besides, LongGen has brought significant inference gain, both in terms of memory and throughput. According to AWS, the largest global cloud provider, inference is estimated to make up 80 to 90% of total ML cloud computing demand [1].
>
> [1] Luccioni, Sasha, Yacine Jernite, and Emma Strubell. "Power hungry processing: Watts driving the cost of AI deployment?." The 2024 ACM Conference on Fairness, Accountability, and Transparency. 2024.
>
> **W2: LongGen with AttnSink and BlockSparse demonstrates similar performance, necessitating a detailed explanation for this observation.**
>
> One possible reason is we keep the same KV cache budget for both methods. We do not perform any hyperparameter search due to the limit of computational resources. Since different methods may require different optimal values for KV cache budget, keeping the same budget may not differentiate the two methods. In the future, we plan to conduct a hyperparameter study for each method given more computational resources.

---

### Official Review · Reviewer_eef1 · 2024-11-03

**Soundness:** 3
**Presentation:** 3
**Contribution:** 2
**Rating:** 8
**Confidence:** 4

**Summary:**

This paper proposes LongGen, which integrates GPU-friendly KV cache reduction architecture to save both length extrapolation and serving cost. It is built on three observations, on sparse attention and the number of tokens needed. It achieves effective cost reduction in both training and serving cost.

**Strengths:**

1. The paper is well written: especially abstract and introduction is well structured and informative on what the paper is going to about. The figures are well made.
2. The performance is very good: e.g. NIAH result is much better than previous methods such as StreamingLLM.

**Weaknesses:**

There is no noticeable weaknesses that the reviewer hope the authors shall address (only some small clarification questions). Please see the question section.

**Questions:**

In the experiment setup, the author mentions that the tensor parallel size is set to 8 with 256 GPUs. Are the remaining GPUs used for data parallelism or pipeline parallelism? And what is the framework used to measured the speedup? And how many iterations to calculate the average training/inference time?

---

> ### Author Response · Authors · 2024-12-01
> **Response**
>
> We thank reviewer eef1 for the valuable time and constructive feedback. Here are our responses to the questions.
>
> **Q1: The author mentions that the tensor parallel size is set to 8 with 256 GPUs in the experiment setup. Are the remaining GPUs used for data parallelism or pipeline parallelism? And what is the framework used to measure the speedup? And how many iterations to calculate the average training/inference time?**
>
> The remaining GPUs are used for data-parallel. Our data_parallel_size = 256/8=32, the same as the global batch size. We found using tensor parallel with checkpoint activation can successfully train a 128k sequence, so we did not use pipeline parallel. To measure speedup, we use the official inference benchmarking script from vLLM. To calculate the average training/inference time, we average over 1000 training steps and 100 inference samples. We found training/inference time are relatively stable and can be estimated using small number of samples.

---

### Official Review · Reviewer_E6wM · 2024-11-04

**Soundness:** 3
**Presentation:** 3
**Contribution:** 3
**Rating:** 6
**Confidence:** 4

**Summary:**

This paper presents LONGGEN, a method designed to enhance the efficiency of long-context training and inference in large language models (LLMs). LONGGEN employs a hybrid attention architecture to achieve an optimal balance between computational efficiency and performance in long-context tasks. The model architecture is segmented into three sections: the first and last thirds of the layers utilize sparse attention mechanisms, while the middle third maintains full attention.

**Strengths:**

1) Problem importance: LONGGEN addresses the critical problem of context window extension in LLMs, specifically targeting issues of performance degradation, high computational complexity, and excessive memory consumption as context length increases, which is an important challenge for enabling LLMs to handle long-form content efficiently in real-world applications.

2) Training and inference efficiency: The LONGGEN approach enhances both training and inference efficiency by reducing training FLOPs, KV cache memory size, prefilling speed, and decoding speed. By employing sparse attention in the outer layers and full attention in the middle, LONGGEN effectively manages computational load and memory usage. This design allows for context length extension up to 128K tokens with lightweight fine-tuning on long-context data, making it suitable for long-context applications.

3) Results: LONGGEN demonstrates almost the same performance as the full-attention model across key benchmarks, including Needle-in-a-Haystack (NIAH), BABILong, RULER, MMLU, Math, and BigBenchHard (BBH). Additionally, it outperforms other KV cache eviction methods and long-context training approaches on the NIAH and BABILong tasks.

**Weaknesses:**

1) Results of the other models: The results of LONGGEN have only been demonstrated on Llama2-7B and Llama2-70B models, which limits understanding of its effectiveness on other model architectures and sizes (such as GPT models, Gemini, or other Llama models).

2) Results on the other tasks: LONGGEN introduces an hourglass architecture that keeps the middle layers in full attention mode, based on previous studies [1, 2, 3] showing that attention heads are crucial for retrieval and reasoning tasks. However, its performance on other long-context benchmarks has not been explored in this work (such as single/multi-document QA or Summarization).

3) Full attention layers:  The specific selection of the full attention layer has not been explored, which can differ from task to task (or model to model). Additionally, although the authors have conducted an ablation study to determine that 1/3 of the layers should use full attention, this proportion may vary across different models or different tasks and would require separate ablation studies for each.

[1] Wenhao Wu, Yizhong Wang, Guangxuan Xiao, Hao Peng, and Yao Fu. Retrieval head mechanistically explains long-context factuality. arXiv preprint arXiv:2404.15574, 2024.

[2] Yao Fu. How do language models put attention weights over long context. Yao FuâA˘Zs Notion ´ , 2024.

[3] Zhenyu Zhang, Ying Sheng, Tianyi Zhou, Tianlong Chen, Lianmin Zheng, Ruisi Cai, Zhao Song, Yuandong Tian, Christopher Ré, Clark W. Barrett, Zhangyang Wang, and Beidi Chen. H2O: heavy-hitter oracle for efficient generative inference of large language models. In Alice Oh, Tristan Naumann, Amir Globerson, Kate Saenko, Moritz Hardt, and Sergey Levine (eds.), Advances in Neural Information Processing Systems 36: Annual Conference on Neural Information Processing Systems 2023, NeurIPS 2023, New Orleans, LA, USA, December 10 - 16, 2023, 2023. URL http://papers.nips.cc/paper_files/paper/2023/hash/6ceefa7b15572587b78ecfcebb2827f8-Abstract-Conference.html.

**Questions:**

1) What are the results of LONGGEN  on other popular large language models, such as GPT models or Gemini?

2) How would the results of LONGGEN change if applied to tasks other than retrieval or reasoning ones?

3) Should the architecture, specifically the number and placement of full attention layers, be adjusted for different models or tasks?

4) Is there an algorithmic or automated method for determining the optimal number and placement of full-attention layers?

---

> ### Author Response · Authors · 2024-12-01
> **Response (1/2)**
>
> We thank reviewer E6wM for the valuable time and constructive feedback. Here are our responses to weaknesses and questions.
>
> **W1: Results of the other models: The results of LONGGEN have only been demonstrated on Llama2-7B and Llama2-70B models, which limits understanding of its effectiveness on other model architectures and sizes (such as GPT models, Gemini, or other Llama models).**
>
> We first explain why we do not test on advanced Llama models, e.g., Llama 3.1 series. We originally planned to test on the Llama-3 series. However, one obstacle is the lack of open-source replication of Llama-3’s pretraining data. Previous studies [1] suggested that during context extension, it is important to keep the same data distribution with pretraining. This usually requires directly up- or down-sampling data from the pretraining corpus. We choose Llama-2 because there is a high-quality open-source replication of its pretraining corpus named Slimpajama [2]. For Llama-3, there is no such corpus. We empirically tried Slimpajama on Llama-3 but observed a significant performance decrease (>10% mmlu) in short context tasks. Therefore, we choose Llama-2 as our model backbone, which shares the same model architecture and training logistics as Llama-3. Besides, using both llama-2 7b and 70b models allows us to test on two attention architectures - 1) conventional multi-head attention and 2) grouped query attention. Using Llama-2 provides a good testbed for this controlled comparison. We plan to extend LongGen to Llama-3 when there is open-source replication of its pretraining corpus.
>
> Second, long context training demands a lot of computing, and our limited computing confines our exploration. Although LongGen only requires lightweight training, it consumes 74 and 387 GPU hours for 7b and 70b models, respectively. Besides, Gemini and GPTs are closed-source models and LongGen could not work without access to model weights and pretraining data.
>
> **W2: Results on the other tasks: LONGGEN introduces an hourglass architecture that keeps the middle layers in full attention mode, based on previous studies [1, 2, 3] showing that attention heads are crucial for retrieval and reasoning tasks. However, its performance on other long-context benchmarks has not been explored in this work (such as single/multi-document QA or Summarization).**
>
> Thank you for the suggestion. We found the maximum context lengths of existing QA and summarization datasets are limited to 16k and 32k. For example, the [SCROLLS](https://www.scrolls-benchmark.com/) benchmark has a cap of 16k tokens. [LongBench](https://arxiv.org/pdf/2308.14508) is also limited to 25k tokens. However, we aim to test our model on 128k context length to validate the effectiveness of sparse training. Aside from needle retrieval, we also test on BABILong and RULER. Two subtasks (subtask 2 and subtask 5) of BABILong require summarizing information from multiple pieces of evidence. RULER also has two QA tasks, which are adapted from SQUAD and HotpotQA.
>
> **W3: Full attention layers: The specific selection of the full attention layer has not been explored, which can differ from task to task (or model to model). Additionally, although the authors have conducted an ablation study to determine that 1/3 of the layers should use full attention, this proportion may vary across different models or different tasks and would require separate ablation studies for each.**
>
> Thank you for the suggestion. In our preliminary experiments exploring various full-layer allocations, we observed that placing the full layers in the middle third consistently delivered strong performance. While it is possible that different tasks may have different optimal full-layer allocations, we chose not to exhaustively investigate this for two main reasons: (1) each variant requires training at least a 128K 7B model, making a comprehensive exploration computationally infeasible, and (2) in practice, long-context models need to handle diverse tasks; therefore, we selected the allocation that performs best overall, even if it may not be optimal for every individual task.

---

> ### Author Response · Authors · 2024-12-01
> **Response (2/2)**
>
> **Q1: What are the results of LONGGEN on other popular large language models, such as GPT models or Gemini?**
>
> Answered in Weakness 1.
>
> **Q2: How would the results of LONGGEN change if applied to tasks other than retrieval or reasoning ones?**
>
> Answered in Weakness 2.
>
> **Q3: Should the architecture, specifically the number and placement of full attention layers, be adjusted for different models or tasks?**
>
> Answered in Weakness 3.
>
> **Q4: Is there an algorithmic or automated method for determining the optimal number and placement of full-attention layers?**
>
> Thanks for the question. It is a good idea. However, as we explained earlier, each search would consume too much computational cost (>360 GPU hours for a 70B model ), and we haven’t done so due to resource limits.
>
> [1] Fu, Yao, et al. "Data engineering for scaling language models to 128k context." arXiv preprint arXiv:2402.10171 (2024).
>
> [2] Shen, Zhiqiang, et al. "Slimpajama-dc: Understanding data combinations for llm training." arXiv preprint arXiv:2309.10818 (2023).

---

> > ### Comment · Reviewer_E6wM · 2024-12-01
> > **Response to authors**
> >
> > Thank you to the authors for their detailed clarifications. After reviewing the responses from the authors and considering feedback from other reviewers, I have decided to maintain my score (slightly above average).

---

### Official Review · Reviewer_Gkd8 · 2024-11-04

**Soundness:** 2
**Presentation:** 3
**Contribution:** 3
**Rating:** 6
**Confidence:** 4

**Summary:**

This paper introduces LongGen, which improves both training and inference efficiency of long-context LLMs. The key insights in this paper are: 1) structured sparse attention patterns, with GPU-friendly memory access patterns, enable practical efficiency gains for long-context LLM; 2) a hybrid architecture with full attention layers and sparse ones provides efficiency merits while keeping the models’ performance; 3) context-extension training only requires a lightweight training dataset. Experiments show that LongGen reduces training wall-clock time by 36%. It also reduces KV cache memory by 62% during inference, achieving 1.67x prefilling speedup and 1.41x decoding speedup.

**Strengths:**

1) LongGen improves both training and inference efficiency for long-context LLMs, accelerating training by 1.55x and inference by 1.41-1.67x, without prominent accuracy loss.
2) LongGen identifies that keeping middle layers with full attention and applying sparse attention on the beginning and final layers achieves better performance. It also finds that keeping 1/3 layers with full attention achieves a balance between accuracy and efficiency.
3) LongGen integrates a triton-based attention kernel supporting structured sparsity for efficient inference and training.

**Weaknesses:**

1) The key insights proposed in this paper have been introduced in other papers. For instance, attention-sink and block-sparse attention are not new attention patterns (e.g., [1](https://arxiv.org/abs/2309.17453), [2](https://arxiv.org/abs/2407.02490)). Additionally, extending the model's context length can be achieved with light-weight training is also observed in existing literature (e.g., [3](https://arxiv.org/abs/2306.15595), [4](https://arxiv.org/abs/2307.03170), [5](https://arxiv.org/abs/2309.12307)). The novelty of this paper is limited.

2) The evaluation details of the efficiency benchmark should be further elaborated.

3) The models used for main evaluation is old. Results on Llama-3 series would be more persuasive.

**Questions:**

1) What is the specific evaluation setting for Figure 2 (Right)? Is it tested on 4 A100 GPUs with TP=4 with vLLM? On a single A100, vLLM (W8A8) requires less than 30ms to decode a token for Llama-2-7B (64K sequence length, batch size = 1). Since the complexity of decoding stage attention grows linearly with regard to the sequence length, decoding a token with 128K context sequence length should take no more than 60ms with a single A100. However, in Figure 2 (Right), it takes 60 seconds to decode 512 token (~117 ms/token) with 4 GPUs (dense baseline with 32 full layers). It would be helpful if the authors can provide more details about this evaluation.

2) What is the kernel-level speedup achieved with the sparsity pattern used in LongGen for the attention kernel? For instance, given the 1/64 sparsity (retain 2K tokens in 128K), what is the speed comparison between dense flash attention and the sparse attention kernel used in LongGen? Is it possible for LongGen to achieve measured speedups when serving (i.e., run inference with) sequences shorter than 128K?

3) How is the proposed method compare with other existing KV cache elimination methods (e.g., [6](https://arxiv.org/abs/2310.01801))?

---

> ### Author Response · Authors · 2024-12-01
> **Response (1/3)**
>
> We thank reviewer Gkd8 for the valuable time and constructive feedback. Here are our responses to weaknesses and questions.
>
> **W1: The key insights proposed in this paper have been introduced. For instance, attention-sink and block-sparse attention are not new attention patterns (e.g., 1, 2). Additionally, extending the model's context length can be achieved with lightweight training is also observed in existing literature (e.g., 3, 4, 5). The novelty of this paper is limited.**
>
> Thanks for pointing out that  KV-reduced attention and lightweight long-context training have received a growing amount of recent interest. This paper aims to answer three research questions that remain unclear in existing works: **1) Whether KV-reduced attention could lead to end-to-end speedup.** This would require the attention pattern to be hardware-aware, supported by a customized training or inference kernel. Besides, some KV-reduced attention methods are incompatible with vllm and TensorRT, the two most popular efficient inference engines. Given this, it is hard for previous KV-reduced attention to bring end-to-end system speedup. To better illustrate this, we compare the throughput of Attention-sink (pytorch-implementation), vllm (full attention), and LongGen (vLLM compatible) in the table below.
>
> | Model                                      | Throughput (token/second) | Improvement |
> |--------------------------------------------|---------------------------|-------------|
> | Attention-sink (pytorch-implementation)    | 23.5                      | x1          |
> | vLLM (full attention)                      | 63.3                      | x2.7        |
> | LongGen (vLLM compatible)                  | 97.6                      | x4.2        |
>
> From the table, we can observe that being vLLM-compatible enables LongGen to improve throughput upon paged attention. **2) Whether KV-reduced attention remains effective in long-context scenarios.** In Figure 3 of the paper, we find directly applying KV-reduced attention to long contexts does not work well. Motivated by the drawbacks, we design LongGen to extend this line of research to long context. **3) Whether lightweight training is still enough for length extension with KV-reduced attention.** Since KV-reduced attention changes the model architecture, it is unclear whether more computing should be put into adapting the model to the new sparse architecture. Our study verifies that 5B tokens suffice the need.
> Therefore, we believe **LongGen is novel and it provides valuable contributions in terms of 1) combining KV-reduced attention and context length extension; 2) a highly efficient training and inference kernel.**
>
> **W2: The evaluation details of the efficiency benchmark should be further elaborated.**
>
> Thanks for the question. For training, we measured the throughput (tokens/second) of training a Llama2-7b model on a 128K sequence length with 256 A100-80G GPUS. We set the tensor parallel size as 8, data parallel size as 32, and use distributed optimizer and checkpoint activation. We do not use pipeline parallel.
>
> For inference, we use the [benchmarking script](https://github.com/vllm-project/vllm/blob/main/benchmarks/benchmark_latency.py) from vLLM. We measure prefilling and decoding latency on one single 128K sequence with a tensor parallel size of 4, using 4 GPUs in total. We use vLLM v0.5.3 with all parameters in bfloat16 format.
> We will explain why it takes 117 ms/token for our dense vLLM baseline in answer to your Question1. We will also add above in the experimental setting of our paper.
>
> **W3: The models used for main evaluation is old. Results on Llama-3 series would be more persuasive.**
>
> Thanks for the great suggestion. We originally planned to test on the Llama-3 series. However, one obstacle is the lack of open-source replication of Llama-3’s pretraining data. Previous studies [1] suggested that during context extension, it is important to keep the same data distribution with pretraining. This usually requires directly up- or down-sampling data from the pretraining corpus. We choose Llama-2 because there is a high-quality open-source replication of its pretraining corpus named Slimpajama [2]. For llama3, there is no such corpus. We empirically tried Slimpajama on llama3 but observed a significant performance decrease (>10% mmlu) in short context tasks. Therefore, we choose Llama-2 as our model backbone, which shares the same model architecture and training logistics as llama-3. Besides, using both llama-2 7b and 70b models allows us to test on two attention architectures - 1) conventional multi-head attention and 2) grouped query attention. Using Llama-2 provides a good testbed for this controlled comparison. We plan to extend LongGen to Llama-3 when there is open-source replication of its pretraining corpus.

---

> > ### Author Response · Authors · 2024-12-01
> > **Response (2/3)**
> >
> > **Q1: What is the specific evaluation setting for Figure 2 (Right)? Is it tested on 4 A100 GPUs with TP=4 with vLLM? On a single A100, vLLM (W8A8) requires less than 30ms to decode a token for Llama-2-7B (64K sequence length, batch size = 1). Since the complexity of decoding stage attention grows linearly with regard to the sequence length, decoding a token with 128K context sequence length should take no more than 60ms with a single A100. However, in Figure 2 (Right), it takes 60 seconds to decode 512 token (~117 ms/token) with 4 GPUs (dense baseline with 32 full layers). It would be helpful if the authors can provide more details about this evaluation.**
> >
> > Thanks for the question. We use the benchmarking script from vLLM. We measure prefilling and decoding latency on one single 128K sequence with a tensor parallel size of 4, using 4 GPUs in total. We use vLLM v0.5.3 with all parameters in bfloat16 format.
> > To explain why it takes 117 ms/token for our dense vLLM baseline -
> >
> > 1) Assuming we only use one GPU and its memory is large enough, we agree that it takes approximately 60ms to decode a token at 128k length, with int8 quantization, which is the vLLM (W8A8) you used. However, we do not do any quantization and instead, load parameters with the original bfloat16 format. This doubles our flops and latency. Therefore, it should take approximately 120ms to decode a token, with a single GPU.
> >
> > 2) A recent [vLLM blog](https://blog.vllm.ai/2024/09/05/perf-update.html) reveals that previous vLLM versions show suboptimal efficiency due to high cpu overhead. Starting from vLLM v0.6.0, it adds asynchronicity among different components and reduces cpu overhead, which results in 2.7x throughput improvement and 5x latency reduction. We used an earlier version of vLLM (v0.5.3) in our setting, which may result in suboptimal efficiency evaluation.
> >
> > 3) We use 4 GPUs instead of 1 since the required memory exceeds the maximum capacity of one single GPU. To explain, 69GB is needed for one 128k llama2-7b sequence. The model weights need at least an additional 14GB. With 4 GPUs and tensor_parallel=4, the computation needed for each GPU is reduced but the communication cost between multiple GPUs is non-negiligible. The time used by NVLink adds another overhead to the theoretical value.
> >
> > **Q2: What is the kernel-level speedup achieved with the sparsity pattern used in LongGen for the attention kernel? For instance, given the 1/64 sparsity (retain 2K tokens in 128K), what is the speed comparison between dense flash attention and the sparse attention kernel used in LongGen? Is it possible for LongGen to achieve measured speedups when serving (i.e., run inference with) sequences shorter than 128K?**
> >
> > First, we illustrate the speed comparison between dense flash attention and the sparse attention kernel (given 1/64 sparsity) used in LongGen. The training speed comparison is illustrated in Figure 2 (left) and inference is illustrated in Figure 2 (right).
> > During training, our chosen “1/3 Full” setting could achieve 1.55 times speedup in total wall-clock training time. Further increasing the sparsity level will bring us more efficiency gain. For example, keeping only 5 full layers (“1/7 Full”)  could result in 1.85 times speedup.
> > Our advantage on inference is more significant. The “32 full layers” setting in Figure 2 (right) refers to the FlashAttention baseline. Compared with FlashAttention, our chosen “1/3 Full (12 Full layers)” setting saves 40% profiling time and 29% decoding time. When only retaining 5 full layers, we could save 54% profiling time and 40% decoding time. Compared with FlashAttention, our training and inference kernel introduce significant efficiency improvement.
> >
> > Second, we show that LongGen could also achieve measured speedups when serving sequences shorter than 128K. We use the same [benchmarking script](https://github.com/vllm-project/vllm/blob/main/benchmarks/benchmark_latency.py) from vLLM as in previous experiments. We evaluate the wall-clock speedup of the sparse attention kernel (given 1/64 sparsity) over FlashAttention and report forward and backward passes separately for clarity. Our comparison is solely focused on the attention kernel and does not include feed-forward and layer-norm.
> >
> > Forward pass latency of one sequence (length ranging from 1k to 128k):
> >
> > |                  | 1k   | 2k   | 4k   | 8k    | 16k   | 32k    | 64k    | 128k    |
> > |------------------|-------|------|------|-------|-------|--------|--------|---------|
> > | LongGen Latency (s)     | 0.12  | 0.21 | 0.39 | 0.86  | 1.99  | 5.40   | 16.67  | 57.07   |
> > | FlashAttention Latency (s) | 0.24  | 0.75 | 2.68 | 10.15 | 39.68 | 156.86 | 645.08 | 2644.51 |
> >
> > From the table, our kernel can cut the forward latency by half even when the sequence length is 1k. The advantage of LongGen grows exponentially by sequence length.

---

> ### Author Response · Authors · 2024-12-01
> **Response (3/3)**
>
> Backward pass of one sequence (length ranging from 1k to 128k):
>
> |                     | 1k    | 2k    | 4k    | 8k     | 16k    | 32k     | 64k      | 128k     |
> |---------------------|-------|-------|-------|--------|--------|---------|----------|----------|
> | LongGen Latency (s) | 0.63  | 1.24  | 2.53  | 5.34   | 12.00  | 29.61   | 83.05    | 256.15   |
> | FlashAttention Latency (s) | 0.94  | 2.68  | 8.64  | 30.56  | 115.82 | 448.78  | 1767.16  | 7015.22  |
>
> From the table, our training kernel consumes less than half the backward time at 2k length. The backward advantage of LongGen also grows exponentially by sequence length.
>
> Again, thank you for the question. We will add the above results to the appendix of our paper.
>
> **Q3: How is the proposed method compare with other existing KV cache elimination methods (e.g., 6)?**
>
> |  | RazorAttention | PyramidKV | FastGen (25%) | FastGen (50%) | LongGen-AttnSink | LongGen-BlockSparse |
> |----------|----------------|-----------|-------|-------|---------|---------|
> |     NIAH Acc     | 46%           | 51%       | 46%   | 55%   | 100%    | 100%    |
>
> As shown in Section 2, Inference-time KV reduction fails to generalize to long contexts. To better answer the question, we additionally provide the needle-in-a-haystack evaluation results of FastGen. We evaluated under 25% and 50% KV cache budget. From the table, FastGen performs similarly with PyramidKV and RazorAttention. We attribute their suboptimal performance to two reasons mentioned in Section 2 - 1) Lack of full context access; and 2) Lack of sparse context adaption.
>
> [1] Fu, Yao, et al. "Data engineering for scaling language models to 128k context." arXiv preprint arXiv:2402.10171 (2024).
>
> [2] Shen, Zhiqiang, et al. "Slimpajama-dc: Understanding data combinations for llm training." arXiv preprint arXiv:2309.10818 (2023).

---

> > ### Comment · Reviewer_Gkd8 · 2024-12-02
> >
> > Thank you for the detailed response. After reviewing the authors' response, most of my concerns have been addressed. I will raise my score to 6. Looking forward to see more results on newer models (e.g., Llama-3 series with GQA).

---

### Official Review · Reviewer_GZk8 · 2024-11-04

**Soundness:** 3
**Presentation:** 3
**Contribution:** 2
**Rating:** 6
**Confidence:** 4

**Summary:**

The paper introduces LONGGEN, an efficient architecture designed to extend context length while minimizing computational and memory overhead during training and inference. The key contribution of LONGGEN is its innovative use of a combination of full attention and KV-reduced attention layers during the post-training phase of context length extension. This approach allows the model to adapt to sparse context scenarios, effectively addressing the poor performance observed with previous KV cache reduction methods on long-context tasks. Additionally, LONGGEN introduces static position access and block-wise context handling to mitigate issues related to position embeddings and idling threads. Empirical experiments demonstrate LONGGEN’s superior performance compared to other KV cache reduction methods, highlighting its effectiveness in managing extended contexts efficiently.

**Strengths:**

Originality
	1. hybrid architecture to long-context expanding. Instead of using a full attention mechanism across all layers, the proposed LONGGEN introduces a creative combination of sparse and full attention. This hybrid method allows model generalize well on long-context tasks without high computational overhead.
	2. The work also introduces an optimized GPU-friendly KV cache management technique, which makes long-context processing feasible and efficient on hardware.

Quality
The paper write with a good formal quality in both methodological and experimental part. Extensive evaluations are presented on benchmarks, highlighting LONGGEN’s performance benefits over alternative KV cache reduction techniques and full-attention baselines. The experimental part also includes ablations analyses such as the position and number of full attention layers.

Clarity
The clarity of the paper is good, particularly in its well explanations of the  KV cache optimization methods and hybrid attention. There are some graphs support the narrative and help show the model’s efficiency benefits.  Clear section headings and figures guide the reader through the technical details, allowing for a smooth understanding of ideas.

Significance
This paper is significant as it addresses a critical challenge in LLM scalability which is efficiently extending the context length while preserving the model’s quality and saving computational cost on hardware. It suitable for some applications that require processing large contexts, such as document analysis and long-form dialogue.

**Weaknesses:**

1. This is not the first paper to propose post-training with a sparse attention mechanism; previous works, such as "Sparser is Faster and Less is More," have also introduced sparse attention methods during both training and inference stages.
2. Comparisons are limited to KV cache reduction methods that allocate KV budgets at the pre-filling stage. However, approaches like "Quest: Query-Aware Sparsity for Efficient Long-Context LLM Inference" use query-level sparsity to dynamically activate the KV cache and provide full context access.
3. Additional experiments are needed to strengthen the robustness of the idea that hybrid sparse attention performs best. For a more comprehensive comparison, LONGGEN could also use H2O, RazorAttention, and PyramidKV methods to extend context length during post-training.

**Questions:**

1. How did you set up experiment parameters to ensure a fair comparison between LONGGEN and previous KV cache reduction methods, such as Attn Sink, H2O, and PyramidKV?
2. Does the sparse attention method remain consistent between training and inference?
3. When you mention savings in profiling and decoding time, is this in comparison to the full attention method? If so, could you also describe how LONGGEN's inference time compares to previous KV cache methods?

---

> ### Author Response · Authors · 2024-12-01
> **Response (1/2)**
>
> We thank reviewer GZk8 for the valuable time and constructive feedback. Here are our responses to weaknesses and questions.
>
> **W1: This is not the first paper to propose post-training with a sparse attention mechanism; previous works, such as "Sparser is Faster and Less is More," have also introduced sparse attention methods during both training and inference stages.**
>
> Thank you for pointing out this related work. We find it very relevant and will add it to our reference. SparseK is available on Jun 24, so we consider it as a concurrent work with a different setting. We think it should not diminish the contribution of LongGen. While SparseK directly trains models on target length, FastGen works in the context length extension (from 2k to 128k) scenario. We combine sparse attention with context extension and verify that lightweight training is enough for length extension with sparse attention.
>
> **W2: Comparisons are limited to KV cache reduction methods that allocate KV budgets at the pre-filling stage. However, approaches like "Quest: Query-Aware Sparsity for Efficient Long-Context LLM Inference" use query-level sparsity to dynamically activate the KV cache and provide full context access.**
>
> Thank you for highlighting this related work. We found Quest is a highly relevant paper and will add them to our baseline comparison experiment. To compare with Quest, we apply it to the same llama2-7b-128k model and measure the accuracy of needle-in-a-haystack retrieval. Note that the setting of our needle retrieval is different from the passkey retrieval in the Quest paper. Our maximum context length is 128k, and our retrieval goal is a fixed piece of fact instead of a random number. During the evaluation, we keep the same sparsity level (KV memory budget) for Quest and LongGen, which corresponds to ~35% of the original KV budget.
>
> |  | RazorAttention | PyramidKV | Quest | LongGen-AttnSink | LongGen-BlockSparse |
> |----------|----------------|-----------|-------|---------|---------|
> |     NIAH Acc     | 46%           | 51%       | 87%   | 100%    | 100%    |
>
> From the table, although Quest performs significantly stronger than prefilling-time KV cache eviction methods, it reaches suboptimal results compared to LongGen. We will conduct more benchmarking experiments in the future and add Quest as a baseline comparison to the paper.
>
> **W3: Additional experiments are needed to strengthen the robustness of the idea that hybrid sparse attention performs best. For a more comprehensive comparison, LONGGEN could also use H2O, RazorAttention, and PyramidKV methods to extend context length during post-training.**
>
> We appreciate the suggestion. One concern of accommodating other patterns is some of them are not hardware-aware. Instead of maintaining static memory access patterns, KV eviction methods (H2O, RazorAttention, and PyramidKV) use dynamic memory management where layout differs in every decoding step. It will incur significant IO overhead and memory fragmentation, which lowers the GPU utilization and decreases the system throughput. Considering it, we prefer to use attention strategies that 1) maintain static access of position and 2) employ block-wise context handling, as explained in Section 3.1.
> Meanwhile, the sparsity pattern in LongGen is highly flexible. Both the number of local blocks and stride length are configurable parameters in our kernel implementation. In the paper, we only demonstrate two configurations due to the resource and page limit, but it would be worthwhile to explore other configurations in the future.
>
> **Q1: How did you set up experiment parameters to ensure a fair comparison between LONGGEN and previous KV cache reduction methods, such as Attn Sink, H2O, and PyramidKV?**
>
> We keep the same KV cache budget for all the compared methods. Specifically, we keep all methods to use 35% KV cache memory of the full attention baseline. For LongGen, this corresponds to 1/3 full attention + 2/3 sparse attention (1/64 sparsity ratio for sparse attention). For other methods, we adjust the hyperparameters that are responsible for sparsity until KV budget approximately equals 35%.
>
> **Q2: Does the sparse attention method remain consistent between training and inference?**
>
> Yes. We use the same architecture with the same hyperparameter settings in training and inference.

---

> > ### Author Response · Authors · 2024-12-01
> > **Response (2/2)**
> >
> > **Q3: When you mention savings in profiling and decoding time, is this in comparison to the full attention method? If so, could you also describe how LONGGEN's inference time compares to previous KV cache methods?**
> >
> > Yes, we compare LongGen with full attention in Figure 2.
> > Previous KV cache methods are mainly implemented based on torch and served without a customized inference kernel. Besides, some KV-reduced attention methods are incompatible with vLLM and TensorRT, the two most popular efficient inference engines. Given this, it is hard for previous KV-reduced attention to bring end-to-end system speedup. To better illustrate this, we compare the throughput of Attention-sink (pytorch-implementation), vllm (full attention), and LongGen (vLLM compatible) in the table below.
> >
> > | Model                                      | Throughput (token/second) | Improvement |
> > |--------------------------------------------|---------------------------|-------------|
> > | Attention-sink (pytorch-implementation)    | 23.5                      | x1          |
> > | vLLM (full attention)                      | 63.3                      | x2.7        |
> > | LongGen (vLLM compatible)                  | 97.6                      | x4.2        |
> >
> > From the table, we can observe that being vLLM-compatible enables LongGen to improve throughput upon paged attention. Among all KV methods (H2O, AttentionSink, PyramidKV, and Quest), Quest stands out by using FlashInfer as its backend. FlashInfer is also an efficient kernel library for serving LLMs. It works in parallel with vLLM. In the future, we plan to conduct fair comparison with Quest by using the same backend.

---

### Meta-Review · Area_Chair_gJNz · 2024-12-23

**Metareview:**

The paper introduces an efficient architecture, LongGen, designed to extend context length while minimizing computational and memory overhead during training and inference. Essentially, LongGen combines full attention and KV-reduced attention layers during the post-training phase of context length extension. This approach allows the model to adapt to sparse context scenarios, effectively addressing the poor performance observed with previous KV cache reduction methods on long-context tasks. Additionally, LONGGEN introduces static position access and block-wise context handling to mitigate issues related to position embeddings and idling threads. Empirical experiments demonstrate LONGGEN’s superior performance compared to other KV cache reduction methods.

Although this paper combines many of the existing techniques, the authors did a solid execution and show decent performance.

**Additional Comments On Reviewer Discussion:**

All reviewers are positive about this paper. Although they raised some concerns, the authors have adequately addressed most of them.

---

### Decision · Program_Chairs · 2025-01-22

Accept (Poster)